# Optimization of the Freezing-Thawing Method for Extracting Phycobiliproteins from *Arthrospira* sp.

**DOI:** 10.3390/molecules25173894

**Published:** 2020-08-26

**Authors:** Hui Teng Tan, Nicholas M. H. Khong, Yam Sim Khaw, Siti Aqlima Ahmad, Fatimah M. Yusoff

**Affiliations:** 1Marine Biotechnology Laboratory, Institute of Bioscience, Universiti Putra Malaysia, Serdang 43400, Selangor Darul Ehsan, Malaysia; huiteng.tan28@gmail.com (H.T.T.); yskhaw@gmail.com (Y.S.K.); 2School of Pharmacy, Monash University Malaysia, Jalan Lagoon Selatan, Bandar Sunway 47500, Selangor Darul Ehsan, Malaysia; nicholas.khong@monash.edu; 3Department of Biochemistry, Faculty of Biotechnology and Biomolecular Sciences, Universiti Putra Malaysia, Serdang 43400, Selangor Darul Ehsan, Malaysia; aqlima@upm.edu.my; 4Department of Aquaculture, Faculty of Agriculture, Universiti Putra Malaysia, Serdang 43400, Selangor Darul Ehsan, Malaysia; 5International Institute of Aquaculture and Aquatic Sciences, Universiti Putra Malaysia, Serdang 43400, Selangor Darul Ehsan, Malaysia

**Keywords:** cyanobacteria, *Arthrospira* sp., total phycobiliproteins, extraction, optimum parameters, freezing–thawing process

## Abstract

The freezing–thawing method had been reported to be the best phycobiliprotein extraction technique. However, optimum parameters of this extraction method for *Arthrospira* sp. (one of the major phycobiliprotein sources) still remained unclear. Hence, this study aimed to optimize the freezing–thawing parameters of phycobiliprotein extraction in *Arthrospira* sp. (UPMC-A0087). The optimization of the freezing–thawing method was conducted using different solvents, biomass/solvent ratios, temperatures, time intervals and freezing–thawing cycles. The extracted phycobiliproteins were quantified using a spectrophotometric assay. Double distilled water (pH 7) with a 0.50% *w*/*v* biomass/solvent ratio was the most efficient solvent in extracting high concentrations and purity of phycobiliproteins from *Arthrospira* sp. In addition, the combination of freezing at −80 °C (2 h) and thawing at 25 °C (24 h) appeared to be the optimum temperature and extraction time to obtain the highest amount of phycobiliproteins. A minimum of one cycle of freezing and thawing was sufficient for extracting high concentrations of phycobiliproteins. The findings from this study could reduce the cost and labor needed for extracting high quality phycobiliproteins. It also allowed the harvesting of large amounts of valuable phycobiliproteins.

## 1. Introduction

Cyanobacteria are prokaryotes that contain chlorophyll *a* to carry out oxygenic photosynthesis, similar to other plants [1]. Besides containing chlorophyll pigments, cyanobacteria also contain phycobilin as an accessory pigment involved in light harvesting. Phycobilin, a water-soluble protein with a wide range of applications has stirred great interest recently as these non-toxic chromoproteins have anti-inflammatory, radical scavenging, antioxidant and hepato-protective properties [2,3,4]. Owing to their numerous benefits, phycobiliproteins have been extensively utilized as colorants in food or used in the pharmaceutical and cosmetics industries [5,6,7] They are classified into three major groups based on their absorption and spectroscopic properties, namely, phycocyanins (PC), phycoerythrins (PE) and allophycocyanins (APC) [6,8]. Among these three types of phycobiliproteins, PC made up to 20% of all the proteins in cyanobacteria [9].

Although cyanobacteria have been proven to be a major source of phycobiliproteins, some major challenges such as suitable cyanobacterial strains, high harvesting cost, and low yield prevented the mass production of phycobiliproteins [10]. Other than the optimization of cultivation conditions that have been studied before, the extraction method plays a vital role in maximizing the recovery of phycobiliproteins from microalgae [11,12,13]. In addition, cell walls of cyanobacteria are multilayered and extremely difficult to disrupt, yet cell rupture is an obligate event to obtain these phycobiliproteins [14]. Furthermore, the diverse compositions and complex cell envelope structures of different cyanobacterial species increase the difficulty of extracting phycobiliproteins [15]. A complete disintegration of the cyanobacterial cells for high pigment yields in a short time is a necessary prerequisite for efficient phycobiliproteins production.

Besides, different cyanobacteria have varying amounts of phycobiliproteins content. *Arthrospira* (commonly known as *Spirulina*) is a genus of the planktonic cyanobacteria that is normally found in alkaline water [16]. It is highly nutritive as it contains up to 10% of lipids, 70% of proteins, 20% of carbohydrates and is rich in minerals, pigments, fibers, as well as vitamins. In addition, this cyanobacterium is reported as being safe for human consumption [17]. Besides, it is one of the filamentous cyanobacteria that contains high amounts of phycobiliproteins, especially phycocyanin (at least 70 mg/g) [16,18,19]. Cifferi [20] also reported the high growth rate of *Arthrospira* sp. with 25% (*w*/*w*) of phycocyanin in its biomass. Therefore, *Arthrospira* sp. as a promising non-toxic source of phycobiliproteins was selected for the present study.

A wide range of extraction methods including homogenization, sonication, lysozyme digestion, heat shock, nitrogen lysis and freezing and thawing (freeze–thaw) have been utilized to obtain phycobiliproteins [21,22,23]. Among these methods, the freezing–thawing extraction method had been reported as being the best for yielding the most phycobiliproteins [24]. However, detailed information on the optimized conditions of the freezing–thawing extraction method is lacking. Hence, the current level of phycobiliproteins production is still relatively low. Therefore, this study aimed to investigate and optimize several parameters of the freezing–thawing method such as types of solvents, pH values, biomass/solvent ratios, temperatures, time intervals and cycles of freezing–thawing for extracting phycobiliproteins from *Arthrospira* sp. (UPMC-A0087). The emphases were on reducing the cost and labor, as well as on enhancing the production rate.

## 2. Results

### 2.1. Double Distilled Water with pH 7 Was Suitable for Total Phycobiliproteins Extraction from Arthrospira sp.

The solvents (double distilled water, DDW; phosphate buffer saline, PBS; sodium phosphate buffer, SPB and potassium phosphate buffer, PPB) in the present study were able to extract at least 10 mg/g total phycobiliproteins from cyanobacteria (Figure 1). Solvent types and their pH were found to contribute significantly to the extraction yield of total phycobiliproteins content (*p* < 0.05). Comparatively, extraction using double distilled water with pH 7 yielded the highest total phycobiliproteins (97.08 ± 8.26 mg/g) from *Arthrospira* sp. compared to the other solvents (*p* < 0.05). In addition, this extraction also produced the highest concentration of phycocyanin, PC (66.39 ± 0.03 mg/g); phycoerythrin, PE (11.98 ± 0.11 mg/g) and allophycocyanin, APC (18.71 ± 0.17 mg/g) among the tested solvents.

The purities of the extracted PC, PE and APC from *Arthrospira* sp. using the solvents in the present study ranged from 0.2–0.85, 0.14–0.40, 0.09–0.44, respectively (Appendix A). Double distilled water with pH 7 extracted the highest purity of phycobiliproteins content (PC: 0.85 ± 0.01; PE: 0.4; APC: 0.44 ± 0.01) compared to the other solvents (*p* < 0.05).

### 2.2. A 0.50% w/v Biomass/Solvent Ratio Was Recommended for Cyanobacterium Arthrospira sp. for Maximum Total Phycobiliproteins

Extractability of total phycobiliprotein content was also significantly affected by the ratio of biomass over solvent used for the extraction (*p* < 0.05). It was observed that the highest biomass/solvent ratio percentage, 4% *w*/*v* (lowest solvent volume: 1 mL), demonstrated the lowest yield of phycobiliproteins (Figure 2). There was neither an obvious increasing nor decreasing trend when using the various biomass/solvent ratio percentages for *Arthrospira* sp. The highest total phycobiliprotein (97.08 ± 0.26 mg/g) extraction from *Arthrospira* sp. was obtained using a 0.50% biomass/solvent ratio. This value was comparable with 2% biomass/solvent ratio with no significant difference (*p* > 0.05). The 0.50% biomass/solvent ratio also extracted the highest yield of PC (66.39 ± 0.03 mg/g) and PE (11.98 ± 0.11 mg/g) (Figure 2). However, the 4% biomass/solvent ratio demonstrated that the extracted phycoerythrin was below the limit of detection. The highest purities of PC (1.94), PE (1.09) and APC (0.77) of *Arthrospira* sp. were obtained using the 4% biomass/solvent ratio (Appendix A).

### 2.3. Optimum Freezing–Thawing Temperature in Extracting Total Phycobiliproteins from Cyanobacteria Was Freezing at −80 °C and Thawing at 25 °C

For the freezing–thawing temperature, less than 10 mg/g of total phycobiliproteins were obtained for *Arthrospira* sp. using freezing–thawing temperatures, which ranged from freezing (F) 0 °C to thawing (T) 4 °C; F 0 °C–T 25 °C; F −30 °C–T 4 °C; F −30 °C–T 25 °C and F −80 °C–T 4 °C with the exception of F −80 °C–T 25 °C (Figure 3). The best freezing–thawing temperature for cyanobacterium *Arthrospira* sp. was found to be freezing at −80 °C and thawing at 25 °C, which provided the highest total phycobiliproteins (97.08 ± 0.26 mg/g) from *Arthrospira* sp. The total phycobiliproteins obtained using this freezing–thawing temperature showed significant differences when compared to the other freezing–thawing temperatures (*p* < 0.05). In addition, this temperature also extracted the highest amount of PC (66.39 ± 0.03 mg/g); PE (11.98 ± 0.11 mg/g); APC (18.71 ± 0.17 mg/g). 

The extract purities of PC, PE and APC of *Arthrospira* sp. ranged from 0.09–0.85, 0.09–0.40 and 0.06–0.44, respectively, using the above-mentioned freezing and thawing temperatures (Appendix A). The highest purity levels of PC (0.85 ± 0.01), PE (0.40) and APC (0.44 ± 0.11) were obtained using the freezing–thawing temperature (F −80 °C–T 25 °C). 

### 2.4. Freezing for 2 h and Thawing for 24 h Were Suggested to Maximize the Extraction of Total Phycobiliproteins from Arthrospira sp.

The treatment of different freezing-thawing time intervals significantly affects the extraction of total phycobiliprotein (*p* < 0.05). Different freezing–thawing time intervals (F 0.5 h–T 1 h; F 0.5 h–T 1.5 h; F 0.5 h–T 2 h and F 1 h–T 2 h) yielded less than 15 mg/g total phycobiliproteins from *Arthrospira* sp. except for the time interval of F 2 h–T 2 h, F 2 h– T 12 h, and F 2 h–T 24 h (Figure 4). The best freezing–thawing interval was freezing for 2 h and thawing for 24 h to extract total phycobiliproteins (219.87 ± 0.68 mg/g) from *Arthrospira* sp. (Figure 4). This time interval demonstrated significant differences in terms of total phycobiliproteins from the other time intervals (*p* < 0.05). Additionally, the highest amounts of PC (172.84 ± 0.37 mg/g), PE (18.14 ± 0.38 mg/g) and APC (28.89 ± 0.50 mg/g) were obtained using this time interval. 

In terms of purities, PC, PE and APC extracts from *Arthrospira* sp. ranged from 0.17–1.95, 0.14–0.50 and 0.08–0.96, respectively (Appendix A). The highest purity levels of PC (1.95), PE (0.50) and APC (0.96) of *Arthrospira* sp. were obtained using freezing for 2 h–thawing for 24 h. 

### 2.5. One Cycle of Freezing and Thawing Was Sufficient to Obtain the Highest Amount of Phycobiliproteins from Arthrospira sp.

Several cycles of the freezing and thawing process on this cyanobacterium provided total phycobiliproteins ranging from 134.38–219.87 mg/g (Figure 5). Notably, one cycle of the freezing and thawing process extracted the highest amount of total phycobiliproteins (219.87 ± 0.68 mg/g) from *Arthrospira* sp. (Figure 5). Extraction of phycobiliprotein with one freezing-thawing cycle showed significantly higher amount of total phycobiliproteins compared to extractions repeated (twice and thrice) freezing-thawing cycle (*p* < 0.05). Extraction using one freezing-thawing cycle also extracted the highest amount of PC (172.84 ± 0.37mg/g); PE (18.14 ± 0.38mg/g); APC (28.89 ± 0.50mg/g). 

The cycles of freezing–thawing demonstrated purities of phycobiliprotein extracts from *Arthrospira* sp., which varied from 0.50–1.95 (PC), 0.15–0.50 (PE) and 0.24–0.96 (APC) (Appendix A). One cycle of freezing–thawing demonstrated the highest purity levels of PC (1.95), PE (0.50) and APC (0.96) from *Arthrospira* sp. 

### 2.6. Amount of Extracted Total Phycobiliproteins from Arthrospira sp. Was Reduced over 24 h

The extracted total phycobiliproteins from *Arthrospira* sp. using the optimum conditions was 219.87 ± 0.68 mg/g. The extracted amounts of PC, PE and APC of *Arthrospira* sp. were 172.84 ± 0.37mg/g, 18.14 ± 0.38 mg/g and 28.89 ± 0.50 mg/g, respectively. These extracted phycobiliproteins were kept at 4 °C for 24 h (Appendix A). 

After 24 h, the extracted total phycobiliproteins from *Arthrospira* sp. was reduced by 6.49% to 205.60 ± 0.94 mg/g. The amounts of extracted PC, PE and APC from *Arthrospira* sp. were decreased over 24 h. The amount of PC was decreased by 4.80% (becoming 164.55 ± 0.17 mg/g), PE was reduced by 15.71% (becoming 15.29 ± 0.49 mg/g) and APC declined by 10.80% (becoming 25.77 ± 0.37 mg/g) after 24 h. 

## 3. Discussion

Numerous researchers have focused on improving the yield of phycobiliproteins either by altering the culture parameters or by subjecting the cyanobacteria to different stress conditions [21,22,25]. Despite these efforts, an efficient extraction method remained a crucial factor in phycobiliprotein production. The freezing–thawing method had been reported to be the most efficient for obtaining phycobiliproteins, particularly PC [26,27,28]. Formation of ice crystals during the freezing process and the rapid thawing process ripped the cell walls of cyanobacteria. The phycobiliproteins would be released from the cells and solubilized in a suitable solvent [24]. In order to obtain the maximum amount of phycobiliproteins, several parameters of the freezing–thawing method such as types of solvent, pH values, biomass/solvent ratios, temperatures, time intervals and cycles are needed to be optimized.

Selection of the solvent is dependent on the polarity and solubility of the target phycobiliproteins [21]. Since phycobiliproteins are water soluble, the use of a polar solvent is effective in extracting them [29]. Double distilled water (pH 7.0) harvested the highest amount and quality of phycobiliproteins from *Arthrospira* sp. among the tested solvents. In previous studies, sodium phosphate buffer (pH 6.8–7.0) had been widely reported as the best solvent in extracting phycobiliproteins from *Spirulina platensis* [22,30]. Silveira et al. [31] reported that the best solvent to harvest PC was sodium phosphate buffer (pH 7.0) in the first 24 h, but thereafter there was no significant difference in extracting PC using sodium phosphate buffer and distilled water. In addition, distilled water extracted the largest amount of PC after 72 h. Moreover, İlter et al. [32] found that extraction using distilled water yielded a two-fold higher amount of PC compared to the phosphate buffer. It was postulated that distilled water might offer similar intracellular conditions such as pH and ionic strength in cyanobacteria [31]. Extraction of phycobiliproteins was also affected by the pH of the solvent. The pH influenced the charge of phycobiliproteins, with enhanced solubility at charged states, while inhibiting solubility at a neutral state [31]. The isoelectric point of PC is between pH 4.74 and 5.8, whereas PE ranges from pH 4.5 to 5.1 [33,34]. Solvents with higher or lower pH from these isoelectric points will increase the solubility of the phycobiliproteins. Extreme solvent pH altered the charge on the protein and led to internal electrostatic attraction. At this point, the protein unwound and the bound solvent was lost, thus resulting in protein denaturation [24]. Several studies had reported pH around 7 was the optimum pH to obtain the maximum amount of phycobiliprotein [22,27]. This was also demonstrated in the present study, with distilled water at pH 7.0 harvesting the highest amount of phycobiliproteins (Figure 1).

Little is known about the biomass/solvent ratio when using the freezing–thawing extraction approach. The highest amount of phycobiliproteins from *Arthrospira* sp. in the present study was extracted using 0.50% biomass/solvent ratio. The maximum PC content (102.97 mg/g) was yielded using the ultrasonication extraction method with 3% biomass/solvent ratio, while a maximum PC concentration (74.74 mg/g) was obtained using the homogenization extraction approach with 2% biomass/solvent ratio [31]. In addition, Pan-utai et al. [35] utilized the ultrasonic-assisted technique with the highest biomass/solvent ratio of 6.67% and obtained a maximum amount of PC (about 60 mg/g) from *Arthrospira platensis*. Silveira et al. [31] reported that PC concentration was strongly influenced by the biomass/solvent ratio and the largest biomass/solvent ratio could provide the highest value of PC. The different biomass/solvent ratios for *Arthrospira* sp. in the present study could be due to the distinct composition of their cell walls. In the case of an excess amount of solvent, absorption of cavitation energy by the solvent from the extraction system occurred causing a lower extraction yield [31]. On the other hand, less amount of solvent might cause incomplete mixing with the cyanobacterial biomass, resulting in low amounts of extracted phycobiliproteins [32]. For *Arthrospira* sp., there was no significant difference in the amounts of phycobiliproteins extracted with usage of 0.50% and 2% biomass/solvent ratios. This could be due to the increased biomass/solvent ratio that might facilitate the diffusivity of the solvent into the cyanobacterial cells leading to higher amounts of extract. Although there was no significant difference found in the extracted phycobiliproteins using 0.50% and 2% biomass/solvent ratios, the 0.50% biomass/solvent ratio was preferred. Firstly, the purities of the extracted phycocyanin, phycoerythrin and allophycocyanin using the 0.50% biomass/solvent ratio (PC: 0.85 ± 0.01; PE: 0.40; APC: 0.44 ± 0.11) were higher than those using 2% biomass/solvent ratio (PC: 0.15 ± 0.02; PE: 0.07 ± 0.01; APC: 0.06 ± 0.01) (Appendix A). Secondly, the losses of the extracted phycobiliproteins content after 24 h using the 0.50% biomass/solvent ratio (16.36%) were less than of those using the 2% biomass/solvent ratio (17.64%).

The freezing–thawing temperature plays one of the crucial roles in the freezing–thawing extraction process. Previous studies utilized freezing temperatures, which ranged from −40 °C to −20 °C and thawing temperatures from 4 °C to room temperature (Appendix A). The optimum temperatures for extraction of the maximum phycobiliproteins content (219.87 ± 0.68 mg/g) from *Arthrospira* sp. in the present study were freezing at −80 °C and thawing at 25 °C. Nevertheless, freezing at around −20 °C and thawing at 4 °C were the common temperatures used to extract phycobiliproteins in previous studies [24,34]. For instance, utilization of this freeze–thaw temperature harvested the maximum content of phycobiliproteins from *Spirulina* sp. (PC: 86.3 ± 1.10 mg/g) [33], *Spirulina platensis* (PC: 146 mg/g) [36], *Anabaena* sp. (total phycobiliprotein (TPB): 128 ± 0.13 mg/g) [24], *Oscillatoria quadripunctulata* (PC: 137.15 mg/g) [34] and *Euhalothece* sp. (PC: 75 mg/g) [37]. On the other hand, freezing in liquid nitrogen and thawing at 4 °C yielded a lower amount and quality of PC [30]. To our knowledge, the freezing at −80 °C and thawing at 25 °C extraction approach in extracting phycobiliproteins had been rarely reported. It is postulated that a large temperature interval of between −80 °C and 25 °C could effectively break the extremely resistant multilayered cell walls of cyanobacteria resulting in higher extracted phycobiliproteins.

The time of freezing and thawing is also one of the factors to determine the yield of extracted phycobiliproteins. The present study demonstrated that freezing for 2 h and thawing for 24 h harvested the largest amount of phycobiliproteins from *Arthrospira* sp. There are limited studies reported about the time interval of freezing and thawing (Appendix A). A minimum of 3 h was used to freeze the cyanobacteria, while a range of 5 min to at least 4 h was utilized to thaw the cells (Appendix A). The time intervals for freezing–thawing were variable among the studies and were highly dependent on the temperatures of freezing–thawing. Sufficient time of freezing is needed to create ice crystal within the cells, while an adequate time of thawing is needed to ensure the complete breakage of the cells in order to obtain the maximum amount of phycobiliproteins. Longer incubation time at room temperature may enhance the phycobiliproteins yield. It is believed that the time of freezing–thawing adopted in this study was ample as the highest phycobiliproteins content was obtained using this time interval. 

The present study demonstrated that one freezing–thawing cycle was adequate to extract the highest amount of phycobiliproteins from *Arthrospira* sp. However, the present finding contrasted with those of several previous studies [18,22,27]. Simis et al. [38] concluded that 5–9 freezing–thawing cycles were necessary for complete extraction of PC. In addition, 3–5 cycles were sufficient to obtain the largest amount of PC from *Cylindrospermopsis raciborskii* [21], *Anabaena spiroides* [21], *Spirulina platensis* [36] and *Spirulina* sp. [39]. These studies utilized the temperature of freezing around −20 °C and thawing around 4 °C room temperature to extract phycobiliproteins. It is hypothesized that the freezing–thawing temperature interval might not be sufficient to disintegrate the multilayer cell walls of cyanobacteria, thus more freezing–thawing cycles were required for these studies. The present finding suggested that the usage of one freezing–thawing cycle could greatly reduce the labor needed and, hence, indirectly increase the extraction efficiency of phycobiliproteins.

Multiple factors (e.g., solvent, biomass/solvent ratio, freeze–thaw temperature, freeze–thaw time interval and freeze–thaw cycle) are the determinants of the phycobiliproteins yield in the freeze–thaw method. Most of the previous studies aimed to extract PC using this method (Appendix A). The highest amount of PC was extracted from *Spirulina platensis* (146 mg/g; purity: 0.87), utilizing sodium phosphate buffer (pH 7), freezing at −20 °C for 3 h and thawing at 4 °C for 5 min and repeating for three cycles [36]. On the other hand, similar extraction conditions, except freezing at −21 °C for 4 h and thawing for 4 h, were applied to *Spirulina platensis*. A lower PC yield was obtained (86.3 mg; purity: 1.34) [33]. Another study utilized phosphate buffer (pH 6.8), 16.67% biomass/solvent ratio, freezing at −40 °C for 4 h and thawing at room temperature for 1 h and repeating for four cycles to extract PC from *Arthrospira platensis*. An even lower PC yield was obtained (73.73 mg/g; purity 0.66) [40]. In the present study, a maximum amount of PC was obtained (172.84 mg/g) using the optimized parameters. This yield could be considered superior compared to those of the previous studies. The optimized conditions in the present study could be considered cost effective. Firstly, the double distilled water was an effective solvent to extract phycobiliproteins in this study, which could lower the solvent cost by approximately eight-fold (www.sigmaaldrich.com). Moreover, water is safe for human consumption. This was supported by Morais et al. [41], who claimed that water was economically viable for phycobiliproteins extraction. Secondly, the optimized conditions only required one freeze–thaw cycle to obtain maximum phycobiliproteins. Though, −80 °C was costly, one cycle could reduce labor and indirectly increase the phycobiliproteins production efficiency. Thus, it could be easily adopted at the industrial level. Previous studies required several cycles to get the phycobiliproteins and the amount obtained was not comparable to that of the present study.

The purity of PC is usually classified based on the absorbance ratio of A_620_/A_280_ [42]. Low purity PC (value of 0.7) is usually utilized by the cosmetic and food industries as bio colorants The C-PC dye can cost between USD 160 and USD 180 per kg [41]. Phycocyanin with purity of 3.9 can be classified as reactive grade. Meanwhile, purity above 4.0 can be categorized as analytical grade. Sekar and Chandramohan [43] reported that the cost of highly purified phycobiliproteins (analytical grade) can reach around USD 5000–30,000 per g. However, the purity of the PE and APC classification still remains unclear. For the sake of simplicity, the purity of PE and APC will be classified according to PC. Overall, the present study successfully extracted food grade phycobiliproteins from *Arthrospira* sp. (except PE). The present findings were consistent with those of other studies [17,27,33]. The purity can be enhanced using suitable purification methods, which have been previously optimized to achieve reactive grade or analytical grade products [44]. 

For stability, the amount of phycobiliproteins (including PC, PE and APC) of *Arthrospira* sp. were reduced over 24 h. However, the reduction was less than 15% except for the PE of *Arthrospira* sp. The present results were in accordance with those of other studies [22,33,42]. Sarada et al. [39] and Doke [33] reporting that extracted PC was more stable for longer periods when kept at 4–10 °C compared to room temperature. Lawrenz and Fedewa [45] demonstrated that extracted PE could be stored at −80 °C for up to 6 weeks without any pigment degradation. However, there is a lack of studies on the stability of extracted APC. Nevertheless, the extracted phycobiliproteins should be stored at 4 °C or at lower temperatures for longer periods of storage.

## 4. Materials and Methods

### 4.1. Cyanobacteria Culture

*Arthrospira* sp. (UPMC-A0087) was isolated using single-celled micro-pipetting and streak plate method from Universiti Putra Malaysia (UPM) lake (2°59′30.3″ N 101°42′57.1″ E). The isolated cyanobacterium was maintained in the Laboratory of Marine Biotechnology (MARSLAB), Institute of Bioscience, UPM. Identity of the cyanobacterium was confirmed using 16s rRNA molecular identification (Genbank accession number: MT490212) and morphological identification [46,47]. *Arthrospira* sp. (UPMC-A0087) was cultivated as xenic culture and grown in a sterilized blue–green medium, BG11, for 28 days [47,48]. The culture was incubated at 25 °C under continuous shaking at 150 rpm and illuminated with fluorescent light of 50 µmol photons m^−2^ s^−1^ following 12:12 light/dark cycle. 

### 4.2. Preparation of Starting Material

The entire extraction process was performed in the dark by wrapping the extraction flasks using aluminum foil. Initially, the cyanobacteria culture at the stationary phase (OD_680_ of *Arthrospira* sp.: approximately 1.00) were centrifuged (Eppendorf, Hamburg, Germany) at 8228× *g* for 10 min. Next, the pellet was washed with double distilled water twice and dried at 40 °C overnight. The dried pellet (approximately 40 mg) was utilized as the starting material. 

### 4.3. Optimization of Freezing–Thawing Extraction Method (Phycobiliprotein Extraction Method)

The optimization of the freezing–thawing method was performed with respect to different solvents, pH values of solvents, biomass solvent ratios, temperatures, time intervals, and number of freezing–thawing cycles. Four different solvents were utilized, namely, sodium phosphate buffer, potassium phosphate buffer, phosphate buffered saline (Sigma-Aldrich, St. Louis, MO, USA) and double distilled water. At the same time, five pH values of the solvents were tested: pH 6, pH 6.5, pH 7.0, pH 7.5 and pH 8.0. As for the biomass solvent ratios, five different ratios were used: 0.25%, 0.50%, 1%, 2% and 4%. Six different freezing–thawing temperature sets were employed, F 0 °C–T 4 °C, F 0 °C–T 25 °C, F −20 °C–T 4 °C, F −20 °C–T 25 °C, F −80 °C–T 4 °C and F −80 °C–T 25 °C, whereas seven different freezing–thawing time sets were applied: F 0.5 h–T 1 h, F 0.5 h–T 1.5 h, F 0.5 h–T 2 h, F 1 h–T 2 h, F 2 h–T 2 h, F 2 h–T 12 h and F 2 h–T 24 h. Three different cycles of freezing–thawing were used. One cycle was defined as the phycobiliproteins content being harvested after one freezing–thawing process at the respective parameters, while two cycles meant the phycobiliproteins content was harvested after two freezing–thawing processes and so forth for the three cycles.

### 4.4. Estimation of Phycobiliproteins Extracted

After undergoing the freezing–thawing extraction method, the mixture was centrifuged at 8228× *g* for 5 min. The supernatant was filtered using a 25 mm cellulose acetate membrane with 0.2 μm pore size (Bonna-Agela Technologies, Wilmington, DE, USA). The absorbance of the supernatants was measured at 562 nm, 620 nm, 652 nm and 750 nm using UV-1900 UV-VIS Spectrophotometer (Shidmazu, Kyoto, Japan). The phycobiliproteins content was quantified based on the following equation given by Bennett and Bogorad [49]. 

Phycocyanin (mg mL^−1^) = [A_620nm_ − 0.474 (A_652nm_)]/5.34

Allophycocyanin (mg mL^−1^) = [A_652nm_ − 0.208 (A_620nm_)]/5.09

Phycoerythrin (mg mL^−1^) = [A_562nm_ − 2.41 (phycocyanin)] − 0.849 (allophycocyanin)/9.62

Absorbances value was corrected for scatter by subtracting the absorbance at 750 nm [25].

### 4.5. Stability of Phycobiliproteins

The first extracted phycobiliproteins were kept at 4 °C. Quantification of phycobiliproteins was repeated after 24 h. The phycobiliproteins content was quantified based on the equation. 

### 4.6. Purity of Phycobiliproteins

The purity of PC was determined by using the ratio A_620nm_/A_280nm_, APC by A_652nm_/A_280nm_ and PE was determined by A_562nm_/A_280nm_.

### 4.7. Statistical Analysis

The collected data were analyzed using one-way analysis of variance (ANOVA), while the significant differences were determined using the Tukey test at 95% confidence interval level. All statistical analyses were performed using the Minitab 17 Statistical Software (Minitab Inc., State College, PA, USA). 

## 5. Conclusions

Both the upstream (strain selection and cultivation) and the downstream (phycobiliproteins extraction process) steps have been gaining interest to be optimized in order to obtain the highest yield of phycobiliproteins [17,23,27]. This study mainly focused on optimizing the conditions of the freezing–thawing method to harvest phycobiliproteins. The best conditions were: double distilled water (pH 7) as solvent, 0.50% biomass/solvent ratio, freezing at −80 °C for 2 h and thawing at 25 °C for 24 h with one freezing–thawing cycle. Double distilled water with an adjustment of pH proved to be able to extract a high yield of phycobiliproteins compared to phosphate buffer, which was commonly used in previous studies. This could reduce the production cost as well as the safety of the extracts. The larger gap of the freezing–thawing temperature interval was believed to be able to extract more phycobiliproteins with just one freezing–thawing cycle. The amount of phycobiliproteins extracted using the optimized conditions of the freezing–thawing process also surpassed the amount that had been reported previously. The present findings could serve as fundamental information in developing a simple, efficient, and cost-effective approach for extracting phycobiliproteins from this cyanobacterium that possibly could be exploited at the commercial level.

## Figures and Tables

**Figure 1 molecules-25-03894-f001:**
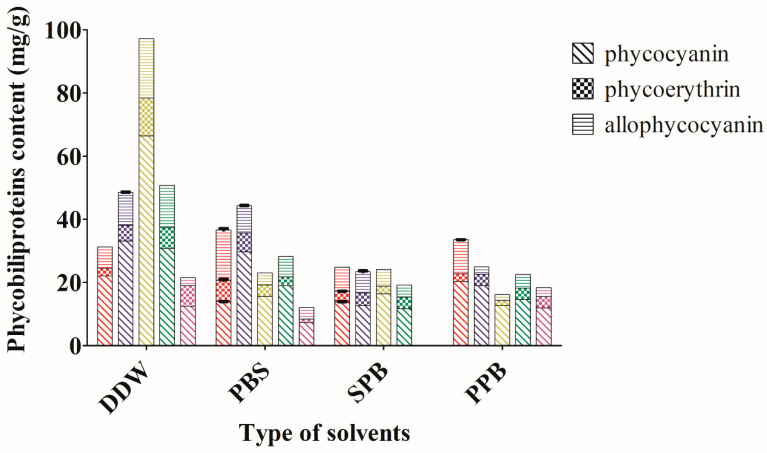
Total phycobiliproteins content extracted from *Arthrospira* sp. (UPMC-A0087) using different solvents with varying pH values. Red color indicates pH 6; blue color indicates pH 6.5; yellow color indicates pH 7; green color indicates pH 7.5; purple color indicates pH 8; DDW indicates double distilled water; PBS indicates phosphate buffer saline; SPB indicates sodium phosphate buffer; PPB indicates potassium phosphate buffer.

**Figure 2 molecules-25-03894-f002:**
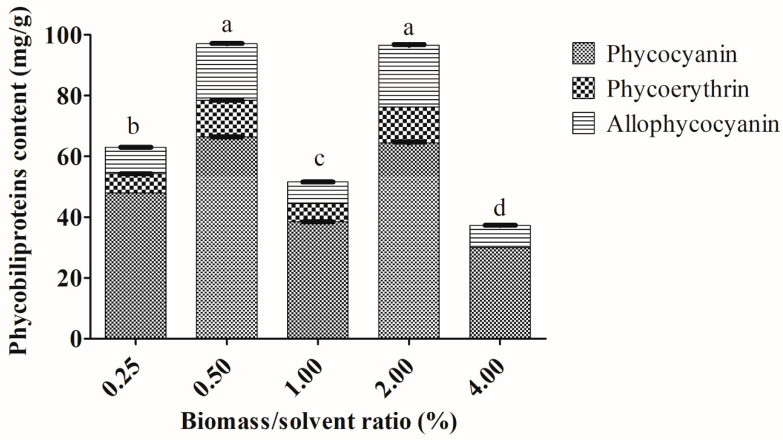
Extractions of phycobiliproteins content from *Arthrospira* sp. (UPMC-A0087) using different biomass solvent ratios. Values annotated with different letters indicate a statistically significant difference (*p* < 0.05) in terms of total phycobiliproteins among the tested parameter.

**Figure 3 molecules-25-03894-f003:**
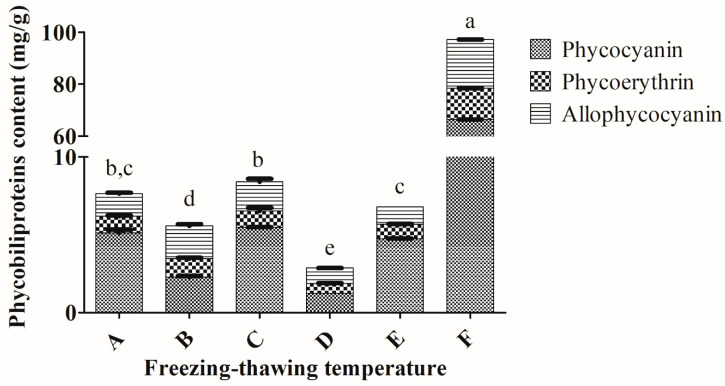
Extractions of phycobiliproteins content of *Arthrospira* sp. (UPMC-A0087) using different freezing–thawing temperatures. A: freezing at 0 °C, thawing at 4 °C; B: freezing at 0 °C, thawing at 25 °C; C: freezing at −30 °C, thawing at 25 °C; D: freezing at −30 °C, thawing at 4 °C; E: freezing at −80 °C, thawing at 4 °C; F: freezing at −80 °C, thawing at 25 °C. Values annotated with different letters indicate a statistically significant difference (*p* < 0.05) in terms of total phycobiliproteins among the tested parameters.

**Figure 4 molecules-25-03894-f004:**
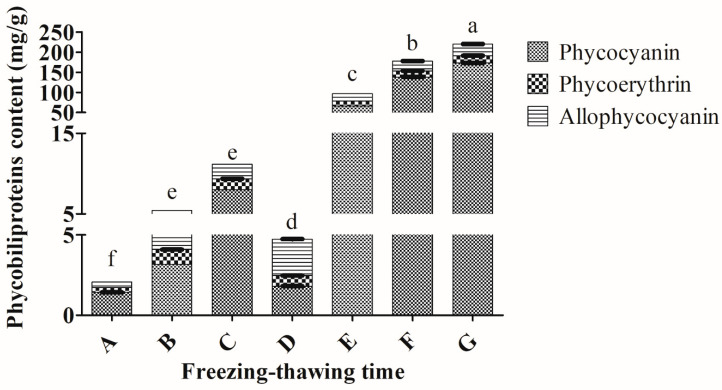
Extractions of phycobiliproteins from *Arthrospira* sp. (UPMC-A0087) using different freezing–thawing times. A: freezing for 0.5 h–thawing for 1 h; B: freezing for 0.5 h–thawing for 1.5 h; C: freezing for 0.5 h–thawing for 2 h; D: freezing for 1 h–thawing for 2 h; E: freezing for 2 h–thawing for 2 h; F: freezing for 2 h–thawing for 12 h; G: freezing for 2 h–thawing for 24 h. Values annotated with different letters indicate a statistically significant difference (*p* < 0.05) in terms of total phycobiliproteins among the tested parameters.

**Figure 5 molecules-25-03894-f005:**
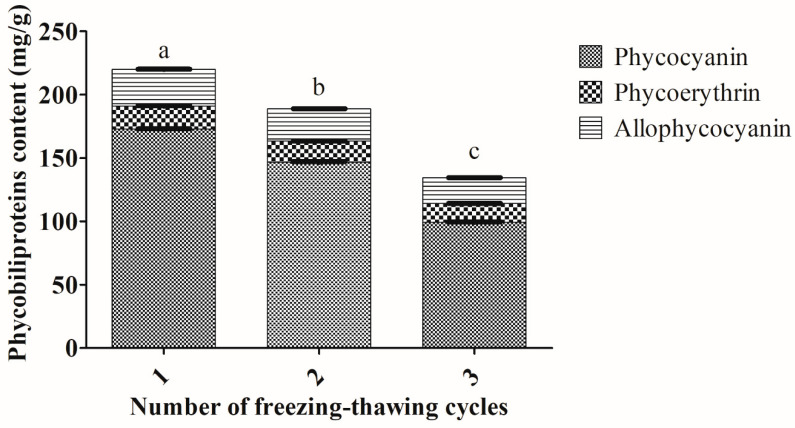
Extractions of phycobiliproteins content from *Arthrospira* sp. (UPMC-A0087) using different number of freezing–thawing cycles. Values annotated with different letters indicate a statistically significant difference (*p* < 0.05) in terms of total phycobiliproteins among the tested parameters.

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
