# Peer review of "Optimization of the Freezing-Thawing Method for Extracting Phycobiliproteins from Arthrospira sp."

_molecules, 2020, doi:10.3390/molecules25173894_

Round 1

Reviewer 1 Report

The authors concluded that the maximum mass of phycobiliproteins from Arthrospira sp. is obtained at a 2% biomass/solvent ratio. But in Fig. 2 has no difference between 0.5% and 2% biomass/solvent ratio, so why such a conclusion? And in general, the results in this figure are difficult to explain. I think it would be good to repeat this study, using the freshly prepared starting material.

Another conclusion of the authors is that "One cycle of freezing and thawing is sufficient to obtain the highest amount of phycobiliprotein". From the results presented in Fig. 5 it can be seen that the largest amount of phycobiliprotein is obtained by combining 2-3 successive extracts.

It would be better to change (in the whole paper) minutes to hours, e.g.

time "of freezing at -80°C for 120 minutes and thawing at 25°C for 1440 minutes"  in minutes should be changed into hours, 2 and 24 h, respectively.

doi for ref. 46 is not correct, it should be doi: 10.1083/jcb.58.2.419

Author Response

Authors’ Responses to Reviewers

Reviewer #1

  1. The authors concluded that the maximum mass of phycobiliproteins from Arthrospira sp. is obtained at a 2% biomass/solvent ratio But in Fig 2 has no difference between 0.5% and 2% biomass/solvent ratio, so why such a conclusion? And in general, the results in this figure are difficult to explain. I think it would be good to repeat this study, using the freshly prepared starting material.

Authors’ responses: Thank you for the comment. Sorry for the mistake. We have made the amendment as follow:

“ 2.2 0.50% biomass/solvent ratio is recommended for….” (page 3, line 99).

“Although there was no significant difference found in the extracted phycobiliproteins using 0.50% and 2% biomass/solvent ratios, the 0.50% biomass/solvent ratio was preferred. Firstly, the purities of the extracted phycocyanin, phycoerythrin and allophycocyanin extracted using the 0.50% biomass/solvent ratio (PC: 0.85 ± 0.01; PE: 0.40; APC: 0.44 ± 0.11) were higher than those using 2% biomass/solvent ratio (PC: 0.15 ± 0.02; PE: 0.07 ± 0.01; APC: 0.06 ± 0.01) (Table S2). Secondly, the losses of the extracted phycobiliproteins content after 24 hours using the 0.50% biomass/solvent ratio (16.36%) were less than of those using the 2% biomass/solvent ratio (17.64%).” (page 10, line 257-264)

The results were shown as table below:

Table 1: Loss of extracted phycobiliproteins content after 24 hours using different biomass/solvent ratio

Biomass solvent ratio (%)

0 h

24 h

Loss of content (%)

0.25

63.02 ± 0.10

60.30 ± 0.33

4.31

0.50

97.08 ± 0.26

81.20 ± 0.21

16.36

1.00

51.56 ± 0.09

44.46 ± 0.03

13.76

2.00

96.54 ± 0.44

79.51 ± 0.57

17.64

4.00

37.31 ± 0.02

33.50 ± 0.05

10.21

*Values represent mean (± standard deviation) of 3 replicates.

*Values reported in mg/g.

We have repeated the experiment using the freshly prepared Arthrospira sp. The repeated results were similar with our previous results (table below). It was found that 0.50% biomass/solvent ratio extracted the highest phycobiliproteins content compared to other ratios.

Table 2: Repeated experiment for biomass solvent ratio

Biomass/solvent ratio

Original results

Repeated results

0.25

63.02 ± 0.10

64.04 ± 0.17

0.50

97.08 ± 0.26

97.34±0.17

1.00

51.56 ± 0.09

78.46±0.14

2.00

96.54 ± 0.44

96.39±0.07

4.00

37.31 ± 0.0210

42.96±0.06

  1. Another conclusion of the authors is that "One cycle of freezing and thawing is sufficient to obtain the highest amount of phycobiliprotein". From the results presented in Fig. 5 it can be seen that the largest amount of phycobiliprotein is obtained by combining 2-3 successive extracts.

Authors’ responses: Thank you for the comment. The cycle in this study means a complete cycle of freezing-thawing process in extracting phycobiliproteins. For instance, one cycle is defined as the phycobiliproteins content was harvested after one freezing-thawing process at respective parameters, while two cycles mean the phycobiliproteins content was harvested after two freezing-thawing process and so forth for three cycles. Since optimum time of freezing is two hours and thawing is 24 hours, the total time needed for 1 cycle of freezing and thawing is 26 hours. Two cycles required 52 hours, whereas three cycles required 78 hours. The highest phycobiliproteins content was obtained after one cycle freezing-thawing process, which is time saving compared to utilization of two and three cycles. Therefore, we concluded that one cycle is sufficient the obtain the highest amount of phycobiliproteins content.

We have added the following text in the manuscript to clarify the cycle of freezing and thawing:

“One cycle was defined as the phycobiliproteins content being harvested after one freezing-thawing process at the respective parameters, while two cycles meant the phycobiliproteins content was harvested after two freezing-thawing processes and so forth for three cycles.” (page 12, line 376-379).

  1. It would be better to change (in the whole paper) minutes to hours, e.g. time "of freezing at -80°C for 120 minutes and thawing at 25°C for 1440 minutes" in minutes should be changed into hours, 2 and 24 h, respectively.

Authors’ responses: We agreed with the reviewer and have made the amendments.

  1. doi for ref. 46 is not correct, it should be doi: 10.1083/jcb.58.2.419

Authors’ responses: Thank you for the comment. We have made the amendment. (page 17, line 565).

Reviewer 2 Report

Ms ID: Molecules-864186

“Optimization of freezing-thawing method in extracting phycobiliproteins from Arthrospira sp.”  Tan and co-workers studied an optimized the phycobiliproteins extraction process in cyanobacteria biomass based on several parameters, such as different solvents used, pH, biomass/solvent ratio, temperature, time interval and freezing-thawing cycles. The subject is very important in the industrial downstream process. The Ms was clearly written and well organized, however this reviewer considers that corrections are needed before publication.

Introduction section: I would like to see more discussion about cyanobacteria of genus Arthrospira in the Introduction section. The authors claimed high production of phycobiliproteins in this genus but they do not discuss about it.

Line 32-33. 4 out of 6 keywords appeared in the manuscript title. Please, choose different keywords that are linked to the main idea of the manuscript to improve the search for this subject.

Lines 54-55: “Arthrospira is a genus…”  instead of  “Arthrospira sp. is a genus…”

Results

Presentation of all graphics should be revised; the external lines of the bars should by thinner. Bars with thinner external lines are easier to compare and error bars will be more apparent.

Line 91: Why did the Authors state: Contrary to expectation,…”?

Line 118, include the minus sign: “…thawing temperature (F -80°C - T 25°C)”.

Line 119: “Freezing for 120 min and thawing for 1400 min…”  instead of  “Freezing at 120 min and thawing at 1400 min…”

Line 165-191. The Authors do not discuss any essential information related with the Ms results. The information presented in these paragraphs that is related with the aims of the study should be in the Introduction section.

Lines 221-222: In addition to biomass/solvent ratio analysis, the authors mentioned that “there was no significant difference in extracting the amount of phycobiliproteins between the usage of 0.5% and 2%”, which might not be true according to Fig. 2. Biomass/solvent ratio of 1% showed a significant decrease of phycobiliproteins extraction when compared to 0.5 and 2% biomass/solvent ratio.

Line 230: “Previous studies utilized…”  instead of   Previous study utilized…”

Line 231: “…temperature from 4 to 25 °C (Table S7).”  instead of  “…temperature at 4 °C (Table S7).”

Lines 229-245: The authors highlighted the importance of freezing-thawing temperatures and time in the extraction process, but there is no comparison with previous studies in the text. The authors should incorporate the information from Table S7 into the Ms to support the discussion. All extraction conditions (Table S7) should be merged throughout the discussion section of the Ms.

Lines 252-254. An option to verify the effect of freezing-thawing cycles on cell wall disintegration is to use Scanning Electron Microscope (SEM).

Lines 257-260: Reference [42], Rito-Palomares et al. (2001) is not the original reference of PC cost. In Rito-Palomares and colleagues study, they mentioned “The commercial value of food grade c-phycocyanin (purity of 0.7, defined as the relationship of 620 nm to 280 nm absorbances) is around $0.13USD per mg, whilst that of reactive grade c-phycocyanin (purity of 3.9) varies from $1 to 5USD per mg”. This information was in accordance to Herrera et al. (1989) [Herrera A, Boussiva S, Napoleone V and Holberg A, Recovery of c-phycocyanin from cyanobacterium Spirulina maxima. J. Appl. Phycol 1:325-331 (1989)].  In addition to high purity PC cost, the $15 USD per mg value was not related with ref. [42] either. This value was found in Prozyme, Catalog of products (1999). None of the above mentioned published studies were acknowledged by the authors. Furthermore, since this information was based on a 1989 article and a catalog published in 1999, the authors should check recent references and update the prices, as appropriate.

Lines 264-265, should be: “…suitable purification methods, which have been previously optimized to…”

Line 271: “…reporting…”  instead of  “…reported…”

Materials and methods

Line 276. Which is the origin of the strain? The authors should mention where the strain came from or if it was bioprospected how the species was identified (molecular biology).

Lines 278-280. The authors should mention how many days the cyanobacteria was cultured.

Line 280. The correct term would be “…12:12 light/dark cycle”.

Conclusion.

Line 328: The authors claimed reduction of cost in the optimized method (and in Abstract, line 65), however according to Figure 3, the optimum freezing-thawing temperature is -80°C, and besides costly, no detailed cost information was provided in the manuscript to assure this cost reduction.

The Conclusion section should be specific according to the results of the Ms. Citation of previous studies should be avoided in this section. Comparison with the results obtained by Hemlata et al. (2011) [19] and Suman et al. (2016) [31] should be incorporated in the discussion section.

English language should be revised carefully throughout the Ms.

Author Response

Authors’ Responses to Reviewers

Reviewer #2

  1. Introduction section: I would like to see more discussion about cyanobacteria of genus Arthrospira in the Introduction section. The authors claimed high production of phycobiliproteins in this genus but they do not discuss about it.

Authors’ responses: Thank you for the comment. We have made the amendments as follow:

“It is highly nutritive as it contains up to 10% of lipids, 70% of proteins, 20% of carbohydrates and is rich in minerals, pigments, fibres as well as vitamins. In addition, this cyanobacterium is reported as being safe for human consumption [19]. Besides, it is one of the filamentous cyanobacteria that contains of high amounts of phycobiliproteins, especially phycocyanin (at least 70 mg/g) [18, 20,21]. Cifferi [22] also reported the high growth rate of Arthrospira sp. with 25% (w/w) of phycocyanin in its biomass.” (page 2, line 60-65)

  1. Line 32-33. 4 out of 6 keywords appeared in the manuscript title. Please, choose different keywords that are linked to the main idea of the manuscript to improve the search for this subject.

Authors’ responses: Thank you for the suggestion and we have amended the keywords in the manuscript as follow:

Keywords: cyanobacteria; Arthrospira sp.; total phycobiliproteins; extraction; optimum parameters; freezing-thawing process (page 1, line 32-33)

  1. Lines 54-55: “Arthrospira is a genus…” instead of “Arthrospira sp. is a genus…”

Authors’ responses: Thank you for the comment. We have revised accordingly as follow:

Arthrospira is a genus…” (page 1, line 34-35)

Results

  1. Presentation of all graphics should be revised; the external lines of the bars should by thinner. Bars with thinner external lines are easier to compare and error bars will be more apparent.

Authors’ responses: Thank you for the suggestion. We have revised the graphics and adjusted the external lines of the bars to become thinner.

  1. Line 91: Why did the Authors state: Contrary to expectation, …”?

Authors’ responses: Thank you for the comment. We have revised accordingly as follow:

“However, …” (page 4, line 108)

  1. Line 118, include the minus sign: “…thawing temperature (F-80°C - T 25°C)”.

Authors’ responses: Thank you for the comment. We have added the minus sign as follow:

“…thawing temperature (F-80°C - T 25°C)” (page 6, line 138)

  1. Line 119: “Freezing for 120 min and thawing for 1400 min…” instead of “Freezing at 120 min and thawing at 1400 min…”

Authors’ responses: Thank you for the comment. We have revised accordingly as follow:

“Freezing for 2 hours and thawing for 24 hours…” (page 6, line 139)

  1. Line 165-191. The Authors do not discuss any essential information related with the Ms results. The information presented in these paragraphs that is related with the aims of the study should be in the Introduction section.

Authors’ responses: Thank you for the comment. We have made the amendments as follow:

“Numerous researchers have focused on improving the yield of phycobiliproteins either by altering the culture parameters or by subjecting the cyanobacteria to different stress conditions [1,13,24–26]. Despite of these efforts, an efficient extraction method remained a crucial factor in phycobiliprotein production. The freezing-thawing method had been reported to be the most efficient for obtaining phycobiliproteins, particularly PC [30,31]. Formation of ice crystals during the freezing process and the rapid thawing process ripped the cell walls of cyanobacteria. The phycobiliproteins would be released from the cells and be solubilized in a suitable solvent [23]. In order to obtain the maximum amount of phycobiliproteins, several parameters of the freezing-thawing method such as types of solvent, pH values, biomass/solvent ratios, temperatures, time intervals and cycles are needed to be optimized.

Selection of the solvent is dependent on the polarity and solubility of the target phycobiliproteins [21]. Since phycobiliproteins are water soluble, the use of a polar solvent is effective in extracting them [33].” (page 9, line 191-217)

  1. Lines 221-222: In addition to biomass/solvent ratio analysis, the authors mentioned that “there was no significant difference in extracting the amount of phycobiliproteins between the usage of 0.5% and 2%”, which might not be true according to Fig. 2. Biomass/solvent ratio of 1% showed a significant decrease of phycobiliproteins extraction when compared to 0.5 and 2% biomass/solvent ratio.

Authors’ responses: We agreed with the reviewer. We have revised accordingly as follow:

“…amounts of phycobiliproteins extracted with usage of…” (page 10, line 253)

  1. Line 230: “Previous studies utilized…” instead of “Previous study utilized…”

Authors’ responses: Thank you for the comment. We have made the amendment as follow:

“Previous studies utilized…” (page 10, line 266)

  1. Line 231: “…temperature from 4 to 25 °C (Table S7).” instead of “…temperature at 4 °C (Table S7).”

Authors’ responses: Thank you for the comment. We have revised accordingly as follow:

“…temperatures from 4°C to room temperature (Table S7).” (page 10, line 267)

  1. Lines 229-245: The authors highlighted the importance of freezing-thawing temperatures and time in the extraction process, but there is no comparison with previous studies in the text. The authors should incorporate the information from Table S7 into the Ms to support the discussion. All extraction conditions (Table S7) should be merged throughout the discussion section of the Ms.

Authors’ responses: We agreed with the reviewer and added the following text into the manuscript:

“The optimum temperatures for extraction of the maximum phycobiliproteins content (219.87 ± 0.68 mg/g) from Arthrospira sp. in the present study were freezing at -80°C and thawing at 25°C. Nevertheless, freezing at around -20°C and thawing at 4°C were the common temperatures used to extract phycobiliproteins in previous studies [23, 37]. For instance, utilization of this freeze-thaw temperature harvested the maximum content of phycobiliproteins from Spirulina sp. (PC: 86.3 ± 0.05 mg/g) [34], Spirulina platensis (PC: 146 mg/g) [32], Anabaena sp. (total phycobiliprotein (TPB): 128 ± 0.13 mg/g) [23], Oscillatoria quadripunctulata (PC: 137.15 mg/g) [40] and Euhalothece sp. (PC: 75 mg/g) [41]. On the other hand, freezing in liquid nitrogen and thawing at 4°C yielded a lower amount and quality of PC [37].” (page 10, line 267-276)

“A minimum of 3 hours was used to freeze the cyanobacteria, while a range of 5 minutes to at least 4 hours was utilized to thaw the cells (Table S7).” (page 10, line 284 - 286) 

“Multiple factors (e.g. solvent, biomass/solvent ratio, freeze-thaw temperature, freeze-thaw time interval and freeze-thaw cycle) are the determinants of the phycobiliproteins yield in the freeze-thaw method. Most of the previous studies aimed to extract PC using this freeze-thaw method (Table S7). The highest amount of PC was extracted from Spirulina platensis (146 mg/g; purity: 0.87), utilizing sodium phosphate buffer (pH 7), freezing at -20°C for 3 hours and thawing at 4°C for 5 minutes and repeating for 3 cycles [32]. On the other hand, similar extraction conditions except freezing at -21°C for 4 hours and thawing for 4 hours were applied to Spirulina platensis. A lower PC yield was obtained (86.3 mg; purity: 1.34) [34]. Another study utilized phosphate buffer (pH 6.8), 16.67% biomass/solvent ratio, freezing at -40°C for 4 hours and thawing at room temperature for 1 hour and repeating for 4 cycles to extract PC from Arthrospira platensis. An even lower PC yield was obtained (73.73 mg/g; purity 0.66) [43]. In the present study, a maximum amount of PC was obtained (172.84 mg/g) using the optimized parameters. This yield could be considered superior compared to those of the previous studies.” (page 11, line 304-316) 

  1. Lines 252-254. An option to verify the effect of freezing-thawing cycles on cell wall disintegration is to use Scanning Electron Microscope (SEM).

Authors’ responses: Thank you for the suggestion. However, the effect of freezing-thawing cycles on cell wall disintegration will be further studied in-depth in the subsequent study. 

  1. Lines 257-260: Reference [42], Rito-Palomares et al. (2001) is not the original reference of PC cost. In Rito-Palomares and colleagues study, they mentioned “The commercial value of food grade c-phycocyanin (purity of 0.7, defined as the relationship of 620 nm to 280 nm absorbances) is around $0.13USD per mg, whilst that of reactive grade c-phycocyanin (purity of 3.9) varies from $1 to 5USD per mg”. This information was in accordance to Herrera et al. (1989) [Herrera A, Boussiva S, Napoleone V and Holberg A, Recovery of c-phycocyanin from cyanobacterium Spirulina maxima. J. Appl. Phycol 1:325-331 (1989)]. In addition to high purity PC cost, the $15 USD per mg value was not related with ref. [42] either. This value was found in Prozyme, Catalog of products (1999). None of the above mentioned published studies were acknowledged by the authors. Furthermore, since this information was based on a 1989 article and a catalog published in 1999, the authors should check recent references and update the prices, as appropriate.

Authors’ responses: Sorry for the mistake, we have changed it accordingly as follow:

“Low purity PC (value of 0.7) is usually utilized by the cosmetic and food industries as bio colorants. The C-PC dye can cost up to $ 160-180 USD per kg [44]. Phycocyanin with purity of 3.9 can be classified as reactive grade. Meanwhile, purity above 4.0 can be categorized as analytical grade. Sekar and Chandramohan [46] reported that the cost of highly purified phycobiliproteins (analytical grade) can reach around $ 5000-30,000 USD per g.” (page 11, line 326-333)

  1. Lines 264-265, should be: “…suitable purification methods, which have been previously optimized to…”

Authors’ responses: Thank you for the comment. We have made the amendment. (page 12, line 337-338)

  1. Line 271: “…reporting…” instead of “…reported…”

Authors’ responses: Thank you for the comment. We have made the amendment. (page 12, line 342)

Materials and methods

  1. Line 276. Which is the origin of the strain? The authors should mention where the strain came from or if it was bioprospected how the species was identified (molecular biology).

Authors’ responses: Thank you for the comment. We have added the following text into the manuscript:

Arthrospira sp. (UPMC-A0087) was isolated using single-celled micro-pipetting and streak plate method from Universiti Putra Malaysia (UPM) lake (2°59'30.3"N 101°42'57.1"E). The isolated cyanobacterium was maintained in the Laboratory of Marine Biotechnology (MARSLAB), Institute of Bioscience, UPM. Identity of the cyanobacterium was confirmed using 16s rRNA molecular identification (accession number: MT490212) and morphological identification [49, 50]” (page 12, line 350-355)

  1. Lines 278-280. The authors should mention how many days the cyanobacteria was cultured.

Authors’ responses: Thank you for the comment. We have added the information. (page 12, line 356)

  1. Line 280. The correct term would be “…12:12 light/dark cycle”

Authors’ responses: Thank you for the comment. We have made the amendment. (page 12, line 358)

Conclusion.

  1. Line 328: The authors claimed reduction of cost in the optimized method (and in Abstract, line 65), however according to Figure 3, the optimum freezing-thawing temperature is -80°C, and besides costly, no detailed cost information was provided in the manuscript to assure this cost reduction.he findings of

Authors’ responses: Thank you for the comment and we have added the following text into the manuscript:

“The optimized conditions in the present study could be considered as cost effective. Firstly,  the double distilled water was an effective solvent to extract phycobiliproteins in this study, which could lower the solvent cost by around 8-fold (www.sigmaaldrich.com). Moreover, water is safe for human consumption. This was supported by Morais et al. 2018 [44], who claimed that water was economically viable for phycobiliproteins extraction. Secondly, the optimized conditions only required one freeze-thaw cycle to obtain maximum phycobiliproteins. Though, -80°C was costly, one cycle could reduce labor and indirectly increased the phycobiliproteins production efficiency. Thus, it could be easily adopted at the industrial level. Previous studies required several cycles to get the phycobiliproteins and the amount obtained was not comparable to that of the present study.” (page 11, line 316-325)     

  1. The Conclusion section should be specific according to the results of the Ms. Citation of previous studies should be avoided in this section. Comparison with the results obtained by Hemlata et al. (2011) [19] and Suman et al. (2016) [31] should be incorporated in the discussion section.

Authors’ responses: We agreed with the reviewer and have made the amendments.  

  1. English language should be revised carefully throughout the Ms.

Authors’ responses: We have sent for English editing. The manuscript has been revised and we hope that it reads better now.

Round 2

Reviewer 2 Report

Dear Editor,

The authors have adequately addressed nearly all my comments and the manuscript has been significantly improved.

I consider that the manuscript deserves publication in Molecules.